# Cephalopods as Challenging and Promising Blue Foods: Structure, Taste, and Culinary Highlights and Applications

**DOI:** 10.3390/foods11172559

**Published:** 2022-08-24

**Authors:** Charlotte Vinther Schmidt, Ole G. Mouritsen

**Affiliations:** Department of Food Science, University of Copenhagen, 26 Rolighedsvej, DK-1958 Frederiksberg, Denmark

**Keywords:** cephalopods, squid, octopus, cuttlefish, texture, taste compounds, umami, gastrophysics, free amino acids, free nucleotides

## Abstract

Foods are complex systems due to their biological origin. Biological materials are soft matter hierarchically structured on all scales from molecules to tissues. The structure reflects the biological constraints of the organism and the function of the tissue. The structural properties influence the texture and hence the mouthfeel of foods prepared from the tissue, and the presence of flavour compounds is similarly determined by biological function. Cephalopods, such as squid, cuttlefish, and octopuses, are notoriously known for having challenging texture due to their muscles being muscular hydrostats with highly cross-linked collagen. Similar with other marine animals such as fish and crustaceans, cephalopods are rich in certain compounds such as free amino acids and free 5′-ribonucleotides that together elicit umami taste. Scientific investigations of culinary applications of cephalopods as foods must therefore involve mechanical studies (texture analysis), physicochemical measurements of thermodynamic properties (protein denaturation), as well as chemical analysis (taste and aroma compounds). The combination of such basic science investigations of food as a soft material along with an exploration of the gastronomic potential has been termed gastrophysics. In this review paper, we reviewed available gastrophysical studies of cephalopod structure, texture, and taste both as raw, soft material and in certain preparations.

## 1. Introduction

We are currently facing a major challenge with the global climate crisis caused by the way we have utilised our natural resources, not least those for food. The global food systems are responsible for about one third of the stresses on the planet [1]. Some of the resources that we extensively use are the marine wild stock reserve of popularly consumed fish species that are suffering from overexploitation, species such as tuna (bigeye, bluefin, yellowfin, and albacore), swordfish, monkfish, Atlantic cod, and salmon [2]. Fisheries have fully exploited 60% of the wild stock and overfished another 30%, and the global catch has declined in recent decades [2]. Aquaculture suffers its problems by its impact on the marine environment and ecological systems. However, access to marine food is essential for population health due to its contents of super-unsaturated fatty acids, essential vitamins, and micronutrients. Moreover, the ocean is also a larder of the so-called *blue food*, e.g., containing protein of animal and algal origin that may constitute a replacement for protein from terrestrial sources, not least land animals. Paradoxically, whereas some marine resources are overexploited or inefficiently used for human sustenance, other bountiful marine food resources are only little used or used to produce feed for terrestrial animals or fish feed for aquaculture. Generally, there is a loss of at least 90% of the nutritional value (e.g., in terms of proteins) at each trophic level of the food chain. Consequently, the application of un- and underexploited blue foods directly as food for human consumption instead will be an ideal way to optimise the way we use our natural global resources while feeding a growing population with nutritious food. It has recently been documented that the environmental performance of blue food is by far superior for farmed bivalves (such as blue mussels) and farmed macroalgae [3].

Among those marine species that are either not used, inefficiently used, or used to feed animals, we find a range of industrial fish (e.g., sprat, sand lance, Norway pout, round goby), macroalgae (seaweeds), bivalves (e.g., blue mussels), echinoderms (e.g., sea cucumbers and starfish), and cephalopods (octopuses, squid, and cuttlefish) [4]. The many reasons for not directly using these species as human food vary depending on convenience, tradition, food culture, and the geographical region. As an example, recent work concerned with the eating potential of these alternative sources of marine protein displays great potential in terms of umami taste in Nordic squid [5], invasive oysters [6], and seaweed [7].

In the present paper, we not only focused on cephalopods, primarily squid, but also drew comparisons with octopus and cuttlefish. Although cephalopods are higher up in the food web than simpler molluscs such as bivalves, their population dynamics indicate that they should be considered as future food with some degree of sustainability. It has been found that all populations of all cephalopod species have globally been on the rise over the last 60 years [8,9]. Hence, an insightful use of these organisms for human food based on the scientific knowledge of how we best can prepare them concerning optimising texture and taste should pave the way for introducing cephalopods to food cultures where they are not commonly consumed [10,11]. Along with this future exploitation of cephalopods as foods are associated ethical issues regarding the treatment and killing of cephalopods [12], not least octopus that clearly, although an invertebrate, is imparted with faculties that may render it both intelligent and possibly possessing consciousness [13].

The present review is based on the existing literature on cephalopod structure, texture, and flavour with a dominant focus on squid. It demonstrates the biological variety of molecular structure within common species of squid, as well as the great differences that exist between species, and it summarises the different methods applied to investigate cephalopod structure, texture, and taste compounds. Additionally, points to be aware of when studying cephalopods’ eating quality are highlighted, including the challenges related to structural variation within specimens as well as the mechanisms of umami taste compound formation and not least the degradation thereof post-slaughter.

The review provides an account of selected culinary investigations of squid and illustrates how a science-based gastrophysics approach can help to form a future cephalopod gastronomy that aims to utilise unexploited but structurally complex biological foods that can be a part of a delicious, nutritious, and sustainable blue-food diet. Seen from a material’s point of view, cephalopod muscle tissue is a unique example of a soft material [14]. Soft materials in the context of gastronomy are best studied by a gastrophysical approach including the use of a wide range of physical and physicochemical methodologies and experimental techniques [15]. Being an emerging scientific discipline [16], such an approach holds a promise for gastronomic and culinary innovation and the development of a cephalopod cuisine.

### Cephalopod Overview

Cephalopods encompass a group within the phylum mollusc and include squid, octopus, and cuttlefish (Figure 1). Cephalopods are found in all salty waters around the globe, and there are about 800 existing species, of which about 30 different species are used as human food [11]. Cephalopods constitute about 5% of the global marine catch, and the volume is increasing. Almost all cephalopods on the world market are from wild stocks, and aquaculture is only very slowly progressing due to difficulties in artificially breeding cephalopods [11].

Within the literature, cephalopods are sometimes referred to by common names and at other times by their Latin name. Within the gastronomical literature, vernacular names are often used, which can lead to confusion regarding the biological identity of the species in question. In the present article, all species will be labelled by their Latin name. Table 1 presents an overview of the species dealt with in the present paper. The selection is determined by which data are available in the literature and which species are of interest in a gastronomical context. Along with the Latin and common names, an indication is given of the typical geographical origin of the species in question.

## 2. Cephalopod Structure by Biological Constraints

In this section, the muscular structure and its variation across cephalopod species are described, with indication of which biological constraints determine the structure. Examples are provided along the way regarding which consequences these relationships have on pre-processing (e.g., handling and freeze-storage effects). For examples regarding the consequences on biological constraints on food processing, the reader is referred to Section 7.2, displaying the available literature on the subject.

### 2.1. Muscular Hydrostatic Mechanics

Terrestrial animals depend on support from the muscles tied to a solid skeletal structure to hold their weight during movement, whereas fish are suspended in a buoyant environment and do not require as resilient and well-connected muscle tissue and can furthermore rely on a lighter bone structure. However, both terrestrial animals and fish mostly work by striated muscles. Molluscs are fundamentally different. Some such as bivalves have an external shell to which the muscles are bound, and their function relies on basically unidirectional contraction [20]. Cephalopods are further different because they have either no or only a rudimentary shell or skeletal structure. Hence, their muscles must be differently organised.

Cephalopods are invertebrates, and, in the case of the families squid, octopus, and cuttlefish, the muscular structure is not supported by an exoskeleton as in the case of shellfish, with a few exceptions of species of cuttlefish. The movement of such coleoid (shell-less) cephalopods is facilitated by the mechanism of muscular hydrostatics, an important biomechanical feature of cephalopods [21,22]. The movement relies on simple rules of physics, where the total volume (and pressure) of the muscular mass is constant, and any decrease in one dimension will consequently cause a compensatory increase in at least one other dimension. Cephalopods thereby move by elongation, bending, and torsion produced by transverse, circular or radial, and longitudinal protein muscle fibres, acting as effectors and support for muscle movement [23]. The same basic principles of support and movement are shared by all structures in coleoid cephalopods with a crucial role played by fibrous collagenous connective tissues [24]. Coleoid cephalopods use ultrastructural modifications rather than tissue-specific myosin isoforms to tune contraction velocities [22]. It has been suggested that the muscle cells share actomyosin composition and sliding mechanisms with those of vertebrate skeletal muscles, but that the control kinetics of the cross-bridge cycle is different [25].

For squid, a specific trait of its structure is the circular smooth muscle fibres in the mantle muscle used for jet propulsion in flight response [26]. As much as 95% of the water inside the mantle can be forced out in one single blow. This requires that the mantle is highly elastic and can withstand a change in the volume of the water-holding cavity by a factor of twenty [10]. This may as well explain the unique character and corresponding challenges to obtaining a familiar and tender piece of muscle tissue after culinary preparation.

### 2.2. Biological Variation

#### 2.2.1. Biological Variation within Specimen

Due to the biological constraints on the muscular structure of cephalopods, a substantial degree of structural variation exists across the mantle of squid (within specimens). Mantle thickness varies along and across the mantle, which in some investigations has been linked to the content of hydroxyproline (Hyp) commonly used as a unique identifying marker for quantifying the content of collagen in tissue [27]. Investigations in squid tissue showed Hyp contents from different parts of the mantle of *L. pealei*, with increasing contents from the top towards the bottom of the mantle, being 58 mg/100 g in the upper third, 65 mg/100 g in the middle third, and 91 mg/100 g in the lower third [28]. Additionally, the content of collagen shows variations in the different layers in the cross-section of the squid mantle [29] (see Section 3.3), revealing a rather advanced structure of the mantle of squid.

Mechanical textural studies of squid using texture analysis (TA) measuring the shear force of different positions of the mantle of *L. forbesii* squid similarly revealed great variations in the required shear force across the mantle, with a toughening effect moving towards the tip of the mantle of *L. forbesii* [30]. Additional studies investigated the mechanical force of the primary (longitudinal) vs. secondary (transverse) axis of squid, i.e., the shear force across and along the circular smooth muscle fibres. Tension tests of the squid mantle texture of *L. pealei* and *I. illecebrosus,* with a focus on how cooking affects the longitudinal vs. transverse axis, found that strength, strain at failure, overall stiffness, and curvature of corrected stress vs. strain differed at different orientations relative to the longitudinal axis of the squid [31] which may be explained by a difference in the collagen content in the longitudinal direction of squid mantle [28]. For the cooked mantle, the mechanical texture is considerably stronger in the longitudinal direction (across circular muscle fibres) than in the transversal direction. Specifically, cooking at 60 °C and 100 °C reveals a greater effect on the mechanical texture of the longitudinal direction than on the transverse, which was attributed to the orientation of the circular collagen fibres of squid [31] and may also additionally be linked to the denaturation of actin fibres [32].

Finally, the influence of biological variability within the specimen is seemingly more significant than the effect of freeze storage as shown in the case of the mantle of squid *I. argentinus* in terms of mechanical texture [33].

It should be stressed that when investigating the texture of the mantle of squid, and possible also *Sepia* and octopus, the position of the investigated cut from the mantle should be noted and possibly precisely tracked due to the biological constraints on the muscular structure of cephalopods.

#### 2.2.2. Biological Variations across Species

Many different species of cephalopods have been investigated in scientific studies in the past decades in relation to cephalopods as food. Common studied species of squid include *Loligo* (*L. vulgaris, L. forbesii, L. pealei, L. duvacelli*), *Illex* (*I. argentinus, I. coindetii, I. illecebrosus*), *Todarodes* (*T. pacificus, T. sagittatus, T. eblanae*), and *Dosidicus gigas*; common studied species of octopus include *Octopus vulgaris* and *Eledone moschata*; and within the group of cuttlefish, the most common species studied in the context of gastronomy is *Sepia officinalis.*

In terms of commonly consumed cuts, it should be noticed that different cuts are used from the different families of cephalopods. Normally, the mantle from squid and cuttlefish is used, whereas it is the arms of octopus that are used, since these are the cuts that comprise the largest part of the animal. However, there are examples in which the mantle from octopus is eaten, and the arms and tentacles of squid are also a popular food in some food cultures. Studies comparing the food uses of the different families of cephalopods should be aware of these differences.

#### 2.2.3. Biological Variation across Cephalopod Families (Squid, Octopus, and Cuttlefish)

In terms of sourcing cephalopods as a future source of marine protein, nutritional quality, food safety, and storage effects are important factors to consider.

Seasonal effects on nutritional quality in terms of protein content, proximate, and fatty acid composition of the body structural tissue have been investigated for various species of cephalopods, such as cuttlefish *S. officinalis,* squid *L. vulgaris,* and octopus *O. vulgaris* and *E. moschata*. The lipid contents are, in general, very low (i.e., lean) for all species, with the lowest lipid content in octopus *E. moschata* (0.60–0.68%) and *L. vulgaris* with the highest level of lipids (1.34–1.92%). For all species, the lipid content varies across the seasons, whereas the protein content does not [34].

The content of essential and non-essential elements from cephalopods (see also Section 6.2) has been compared for octopus, cuttlefish, and squid. Total concentrations of essential (Cu, Zn, Se, and Cr) and non-essential (Hg, Cd, Pb, and As) trace elements in the tissue and hepatopancreas of Octopodidae (*E. moschata*, *E. cirrhosa*, *O. salutii*), Sepiidae (*S. elegans*, *S. orbignyana*) and Loliginidae (*I. coindeti*, *L. vulgaris*) greatly vary across cephalopod families. The highest toxic element concentrations have been found in Octopodidae (Hg: 0.44; Cd: 0.49; Pb: 0.10 mg g^−1^ wet weight) and Sepiidae (Hg: 0.27; Cd: 0.50; Pb: 0.12 mg g^−1^ wet weight), while Loliginidae species contain less of the health-affecting elements, especially Hg (Hg: 0.11; Cd: 0.30; Pb: 0.05 mg g^−1^ wet weight). The other elements show a heterogeneous distribution among the different cephalopod families [35]. It should be noted that the actual values of the contents of these elements, not least the toxic elements, will be dependent on the geographical location and feeding status of the actual specimens investigated.

Effects of freezing at −45 °C and freeze storage at −18 °C over a period of 30 days on protein functionality and texture of cephalopod muscles have been investigated using TA, sensory evaluation, total protein, and free amino acid (FAA) analysis, as well as water holding capacity [36]. The mechanical texture of squid and cuttlefish is significantly affected after freeze storage for 30 days, but not that of octopus. Whereas the mechanical hardness of squid significantly decreases, the hardness of cuttlefish increases (but not significantly) although chewiness significantly increases. However, the corresponding sensory evaluation of hardness and chewiness found increasing effects in all three species of cephalopods, with the least increase for squid and the largest for cuttlefish [36].

In conclusion, based on these data, squid compared with octopus and cuttlefish may be favourable to utilise culinary, considering stable textural behaviour during freeze storage [36], the highest amount of healthy mono- and polyunsaturated fatty acids [34], and the lowest amounts of health-affecting trace metals [35].

##### Squid vs. Octopus

The protein structure of the musculature of coleoid cephalopod arms and tentacles differs between squid arms and tentacles and between squid and octopus arms [22]. In a review article on collagen in cephalopods and its physicochemical properties and influence on muscle texture, different amounts of collagen components in different species of cephalopods were found. It was found that there are similar amounts of Hyp in the arms of squid (*D. gigas*) and octopus (*O. vulgaris*), whereas the content of Hyp in octopus *E. cirrhosa* is almost twice as high [37].

##### Squid vs. Cuttlefish

Investigations into collagen can reveal both the quantity (i.e., the content of Hyp) as well as the quality (i.e., the degree of cross-linking and collagen solubility). In a study investigating the relationship between heat-induced collagen solubility and the content of pyridinoline (Pyr), a cross-linking compound of collagen fibres, in squid mantle, the collagen in five species of squid was compared with that in one species of cuttlefish. Overall, a large amount of collagen (45–70%) was found not to be solubilised even after prolonged heating, indicating a larger portion of cross-linked collagen. The least amount of insoluble collagen was found among squid, *L. pealei* and *T. pacificus*, and the highest amount in cuttlefish and flying squid [38]. The solubility of collagen varies among different species of squid with examples having both less and more insoluble collagen compared with cuttlefish.

In another study of textural quality indicators of cephalopods, squid (*Photololigo duvaucelii*) and cuttlefish (*S. aculeata*) stored on ice were compared. From this study, it was shown that acid-soluble peptides, ammonia content, and expressible drip loss significantly increase with increasing storage time. In terms of textural effects, a significant decrease in shear force and sensory texture appears for both squid and cuttlefish [39]. Thus, compared with the findings from [36] (Section 2.2.3), there are some indications that freezing will tenderise squid at temperatures around and well below 0 °C. This may be different for cuttlefish, where a tenderising effect applies only when stored at temperatures around 0 °C, while the opposite occurs when stored at temperatures well below the freezing point (−18 °C) or as a consequence of the freezing process itself (in the given study at −45 °C, [36]).

Seemingly, collagen quantity and quality cannot simply be linked to the specific cephalopod family. However, freeze-storage effects may cause different effects in different cephalopod families, and caution must be exercised for the scientific studies of cephalopod texture when comparing different families.

## 3. Molecular Structure of Cephalopods

Cephalopod structure has been investigated by multiple techniques, some giving direct access to structural information such as microscopy and scattering techniques, and others more indirectly such as spectroscopy as well as chemical and thermal analysis. Some relevant studies include electrophoresis [38,40,41,42,43,44,45], microscopy [29,32,42,43,46,47,48,49,50,51,52], spectroscopy [41,53], small-angle x-ray scattering [32], differential scanning calorimetry [32,43,54], and chemical analysis in relation to collagen quantity and quality determination [28,38,40,53,54,55,56,57,58,59]. The reader is referred to the literature for a detailed description, and the following section is limited to highlighting results obtained from the most common techniques that include microscopy, spectroscopy, and chemical analysis.

### 3.1. Microscopy

Based on scanning electron microscopy, five major layers can be identified in the mantle of squid (*L. pealei*) [29]. The first layer from inside the cavity of the mantle is a visceral lining. The next layer is an inner tunic, consisting of fibrous aggregates loosely bound and interwoven in a mesh. In the middle layer, comprising 98% of the mantle thickness, a sarcoplasmic core of muscle fibres in circumflexial bands is found in a row with occasionally orthogonal bands connecting the inner and outer tunics. Next is an outer tunic consisting of layers of collagenous fibres in a chicken-wire-mesh structure. Finally, the last layer is an outer lining consisting of randomly sized and ordered fibres of connective tissue [29].

Other types of microscopy have also been applied to study the structure of squid and octopus; some examples include second-harmonic-generation microscopy (SHGM) applied to investigate the structural changes inside the mantle caused by low-temperature long-time cooking [32]. For *L. forbesii*, the results from SHGM linked to the investigations by TA, differential scanning calorimetry (DSC), as well as cooking-loss measures, and the major responsible layers controlling texture were identified as collagen and actin, with an ideal cooking temperature to be around 55 °C, while being >46 °C but <77 °C. Cooking loss was linked to the denaturation of collagen (and, to some degree, myosin), whereas mechanical texture was, in general, linked to collagen and actin, with visual imaging from SHGM validating the results from DSC, suggesting that controlling the denaturation of collagen is key for obtaining both tenderness and juiciness of squid mantle [32]. Microscopy, using light microscopy and transmission electron microscopy combined with TA, was also used to investigate refrigeration effects. Changes in the firmness and structure of the mantle of *L. bleekert* during refrigeration (5 °C) were also studied. It was shown that the shear force of raw mantle sharply decreased after between 3 and 9 hrs of refrigeration after killing, with light microscopy observations revealing many pores between muscle cells in the softened raw samples, which was suggested to be a result of the detachment of muscle cells from connective tissues [48].

Studies into cephalopod structure using microscopy further investigated the impact of drying methods (freeze-drying vs. heat pump) [42,43], enzymatic tenderisation [46], deep frying [47], soaking post cooking and drying [49,50,51], as well as low-temperature long-time cooking [32,52].

### 3.2. Spectroscopy

Spectroscopic investigations of cephalopods often involve the detection of Pyr, a fluorescent cross-linking compound of collagen formed in matured collagen. In a study comparing methods against each other, it was argued that ^1^H-NMR and Raman spectrometry provide an assay that is more sensitive than fluorescence and HPLC for Pyr detection in squid tissues [53].

Spectroscopy can also be used for other purposes. The effects of different drying methods on the myosin structure, amino acid composition, protein digestibility, and volatile compound profile of *T. pacificus* were studied by ultraviolet circular dichroism spectroscopy [41]. This study found changes in the secondary and tertiary structure of the proteins, and by two-dimensional gas chromatography with a time-of-flight mass spectrometric detector (GC × GC–TOFMS), the study was, furthermore, able to detect a decrease in the total number of volatile compounds and an increase in the total content of volatile compounds.

### 3.3. Chemical Analysis

Below we review some aspects of the chemical analysis of cephalopods that are of particular relevance for cephalopod gastronomy, specifically collagen structure that is key to texture and amino acid contents that are important for taste, particularly umami (see also Section 5.1.1).

#### 3.3.1. Collagen

A determining factor for tough and rubbery animal tissue quality is the content and quality of collagen. It has been suggested that the texture of cephalopods is tougher than that of other marine organisms because their muscle fibres are covered by more extensively cross-linked connective tissue, making it more stable during post-catch handling, including heating, freezing, and other processes that affect the quality of the tissue of the meat [37].

The collagen content in Nordic *L. forbesii* was analysed for its content of Hyp by liquid chromatography and was estimated to contain the highest amount in the arms (97.0 mg/100 g) and then fins (68.8 mg/100 g) and the lowest amount in the mantle (37.6 mg/100 g), respectively, corresponding to 4.3, 3.0, and 1.5% collagen of total protein [54]. For Indian squid *L. duvauceli* Orbigny, a somewhat higher value of hydroxyproline was reported to be 140 mg/100 g in the mantle and 150 mg/100 g in arms, corresponding to 12.1% collagen of total protein in the mantle [59]. Other authors reported lower values of collagen content, in the range of 3% to 11% (of total protein) in the mantle of *Illex* and *Loligo* [58]. Higher values were reported for the mantle of *D. gigas*, consisting of as much as 15.0% collagen [57].

The collagen content has also been analysed in the different layers across the mantle of squid. The content of Hyp in the different layers of the mantle of *L. pealei* was found to be 62 mg/100 g in the muscle fibres, 107 mg/100 g in the inner tunic, and 270 mg/100 g in the outer tunic, each layer comprising 98–98.5%, 0.1–0.5%, and 0.5–1.5% Hyp, respectively [28].

The degree of collagen cross-linking is a major factor in the perceived texture of cephalopods. Collagen cross-linkages can be enzymatically or non-enzymatically formed and are generated from the presence of the amino acids lysine (Lys) and hydroxylysine (Hyl). A common cross-linking compound in mature collagen is Pyr formed by lysyl oxidase, which is the most common route of cross-link formation. The non-enzymatic route will mainly occur from glycation [60].

The nature of cross-linking can be assessed from chemical analysis coupled with thermodynamic measurements of the denaturation temperature of the insoluble collagen fraction. In some cephalopod species, squid *D. gigas* and octopus *O. vulgaris*, the denaturation temperature of the insoluble collagen fraction is found to be related to the Pyr content [37]. Pyr is also closely related to collagen solubility in the mantle of different squids along with the high thermal resistance of the muscle during heating [38]. For octopus, a relationship between mechanical textural shear force, Pyr concentration, and lysyl oxidase has been found (*Octopus vulgaris*) [37].

#### 3.3.2. Amino Acids

The analysis of those amino acids in their free form that are important for taste, specifically umami, is described in Section 5.1.1, and their degradation during food processing is described in Section 5.2.

In a study of freeze-dried *T. pacificus*, various drying techniques were investigated to reveal their potential damaging effect on the total amino acid content. It was shown that a similar total amino acid profile as that of raw squid was present after freeze drying. In particular, the amino acid compositions of hot-air-dried and heat-pump-dried squids were similar, whereas the in vitro digestibility of heat-pump-dried squid was significantly higher than that of hot-air-dried squid. Hot-air drying turned out to lead to more damage to the myosin structure than heat-pump drying, while freeze drying effectively retained the myosin integrity [41].

## 4. Cephalopod Texture

The structural properties of cephalopod muscular tissue as described above determine the textural properties of the tissue considered as food [61]. Texture is the sensory perception of the elements of food structure and as such intimately dependent on the hierarchical and multiscale structuring of the food matrix [62] as a piece of complex soft matter [14]. In many ways, the cephalopod muscle is one of the most intriguing pieces of soft material used for food [11].

### Texture Analysis

Cephalopod structure has been extensively investigated scientifically since the 1970s using the texture analysis (TA) methodologies of food pioneered in the 1960s by Dr Alina Surmacka Szczesniak (2002). Several authors have investigated squid and octopus texture by TA to study multiple effects. Historically, the TA of cephalopods has included methods including compression or puncture test by a cylindrical probe, shear force by a knife or razor blade, and collagen gel strength test. Sample cutting and orientation of muscle tissue samples analysed by TA highly affect the outcome. Specifically, in the case of squid, the direction of the shearing and sampling area of the mantle affects the TA due to the biological structural variation across the mantle thickness [28] and along the longitudinal direction [31,33] and due to the direction of dominant circular muscle fibres in the mantle [28,31,49,63].

Table 2 provides an overview of the studies dealing with TA of the textural properties of cephalopods. With regard to studies of textural changes of cephalopods caused by cold and freeze storage, see Section 2.2.3. For other specific results, the reader is referred to the quoted literature.

## 5. Flavour of Cephalopods

The flavour of cephalopods as foods has contributions from both water-soluble taste compounds and volatile aroma compounds. The flavour of cephalopods has been studied in case of aroma and basic taste properties (sweet, sour, bitter, salt, and umami). Particular attention has been paid to those compounds that elicit umami taste and enter in umami taste synergy [30,83], specifically certain free amino acids (glutamate and aspartate) and free 5′-ribonucleotides.

Some flavour compounds are produced *post mortem* by various enzymatic and microbial degradation processes, whereas others may be formed as a consequence of preparation processes. The details of the flavour profile will depend on the species in question as well as the feeding status and habitat (merroir) of the species. Still, being molluscs living in salty waters, cephalopods have some general flavour characteristics shared with other marine organisms such as the saltiness and flavour of the osmolytes that are needed to counterbalance the osmotic pressure in salty water. Such osmolytes include free amino acids and trimethylaminoxid (TMAO). It is hypothesised that cephalopods use TMAO as an osmolyte to a larger extent and sweet-tasting amino acids (e.g., glycine and alanine) to a lesser extent than, for example, mussels [10]. However, this matter is not completely as clear for cephalopods as it is for sharks [84]. Nevertheless, TMAO is tasteless, and cephalopods, therefore, is thought to have a less sweet taste than other molluscs. *Post mortem*, the TMAO is transformed by endogenous enzymes into trimethylamine (TMA) that has an unpleasant fishy odour. Due to the higher content of TMAO in cephalopods compared with, for example, fish, they are more prone to develop a fishy aroma if not kept at a low temperature [10].

### 5.1. Taste Components of Squid

#### 5.1.1. Umami Taste Compounds

Free amino acid (FAA) analysis by high-pressure liquid chromatography (HPLC) or gas chromatography–mass spectroscopy (GC-MS) are common methods to quantify amino acids. Before amino acid analysis, the sample needs to be stabilised and water has to be removed. In the context of squid, it has been shown that freeze drying followed by heat-pump and hot-air drying retained the highest protein quality and digestibility [41]. The squid mantle of *L. forbesii* has been found to contain 110 mg/100 g of free glutamate [5], which is within the range of that found in mussel (110 mg/100 g), shrimps (120 mg/100 g), and scallops (140 mg/100 g) and substantially higher than that found in pork (10 mg/100 g), chicken (20–50 mg/100 g), and beef (10 mg/100 g) [5,85]. Other studies reported contents of glutamate levels ranging from 4 mg/100 g in *Sepioteuthis lessoniana* [86] to 110 mg/100 g in *Loligo forbesii* [5], as summarised in Table 3.

In terms of 5′-ribonucletides, the contribution from 5′-ribonucleotides (AMP, IMP, XMP, GMP) that can enter umami synergy with free amino acids (in particular glutamate) can be evenly compared in terms of IMP equivalence using the conversion factors proposed by Yamaguchi (1971) [87]. Consequently, the content in *L. forbesii* is 24 mg/100 g IMP equivalence, which is similar to that found in sundried tomatoes (23 mg/100 g IMP, based on 10 mg/100 g GMP) but lower than that in shrimp (90 mg/100 g) and substantially lower than that in mackerel (130–280 mg/100 g) [85].

The available data for the contents of umami compounds in cephalopods are somewhat scarce, and it is mostly results from squid that have been reported as illustrated in Table 3. For the sake of comparison, this table also contains data for other selected marine species. In this table, the effective umami concentration (EUC) is also given as calculated from the classical formula [87].

From Table 3 it is worth noting that despite scoring much lower than kelp seaweed, *L. forbesii* together with *I. argentinus* display the highest concentrations of Glu, which is within the same range as that in clams and scallop and much higher than that in shrimp, lobster, crab, and eel. The remaining species of squid are within the range of shrimp, lobster, crab, and eel in terms of the content of free Glu. In terms of nucleotides, squid contains nucleotides within the same concentration range as the other marine species listed in Table 3, only surpassed by tuna and oyster species *Ostrea edulis*. To the extent of the knowledge of the authors, no data were available for free amino acids and 5′-ribonucleotides for cuttlefish, and only one source for octopus was found. Umami taste compounds of octopus and cuttlefish deserve to be investigated in future studies of cephalopods.

#### 5.1.2. Other Taste Compounds

Other taste compounds related to cephalopod flavour are NaCl, succinic acid, octopine, lactate, amino acids, and 5′-ribonucleotides. Using a series of omission and addition tests and sensory evaluation of synthetic extracts, the taste-active components of the mantle of *S. lessoniana* were determined to be glycine, alanine, proline, glutamate, arginine, adenosine 5-monophosphate, trimethylamine oxide, and glycine betaine, as well as potassium, sodium, and chloride ions [86]. According to sensory evaluation, some taste compounds may be more important for the squid taste than others. In the same study [86], a simplified synthetic extract prepared from just 11 taste-active components was found to almost simulate the taste of the complete synthetic extract of 45 components, being the case for *Sepioteuthis lessoniana* as well as *L. bleekeri, L. edulis*, and *Todarodes pacificus*. Umami components (Glu 4 mg/100 mL and AMP 249 mg/100 mL) were, in this case, not recognised as a major taste attribute of these species in the sensory omission test.

### 5.2. Degradation Processes of Umami Taste Compounds in Cephalopods

Decomposition kinetics of chemical taste compounds, especially those of taste-active amino acids and nucleotides, is highly relevant for storage and food preparation concerning food quality.

In fish muscles, free 5′-ribonucleotides are generated from the remaining adenosine triphosphate (ATP) in the muscle tissue. *Post mortem* ATP and ADP (adenosine diphosphate) are converted into umami-tasting IMP, which is further degraded into bitter-tasting hypoxanthine and inosine by enzymatic processes [100].

For cephalopods, *post mortem* changes of ATP and ATP metabolites in *D. bleekeri* have been investigated during storage. AMP peaked after 1.5 days, IMP peaked after 1.5 days in the arms and fin, while IMP peaked in the mantle after 2–4 days. It was further noticed that the content of ATP metabolites was much higher in the mantle than in the arms and fins (9 vs. 5 μmol/g) [101]. Another study investigating *Dosidicus gigas* found that AMP slightly increased up to approximately 3 days on ice storage, after which it decreased [102]. For similar investigations of storage effects on beef, chicken, and pork, Glu is found to increase while AMP and IMP are found to decrease [103,104], an effect that also applies to fish [105].

Degradation of taste-active amino acids and nucleotides during the food processing of cephalopods is more rarely investigated. Nonetheless, degradation of Glu and ATP metabolites has been found in squid during sous vide cooking at low temperature for a long time [5], degradation in *O. bartrami* and *I. argentinus* during the processing of dried-seasoned squid [92], and degradation in *T. pacificus* after high-pressure processing (HPP) [89]. Sous vide cooking also showed a slight decrease in the content of free Glu from 110 to 99 mg/100 g, and on average an overall unchanged content of GMP, IMP, and XMP, whereas the content of AMP increased from 48 to 56 mg/100 g [5]. In another study, processing of dried-seasoned squid facilitated a significant increase in Glu in the case of *Illex* (not pre-boiled), with a significant increase after both semi-drying as well as after subsequent roasting. The semi-drying step further significantly increased the content of IMP. A similar effect was not found for Ommastrephes (pre-boiled), which had a lower content of Glu after the same processing steps compared with that in raw tissue. As the pre-boiling step differed between the two species of squid, it is, however, not possible to discern whether the changes in chemical composition were caused by the pre-boiling or related to the nature of the species [92]. For other types of foods, the same seems to apply: for eels during different processing steps, sterilisation significantly decreases the FAA content of Glu [99].

To summarise, it may seem that free amino acids are prone to degrade with harsh processing, whereas they may increase during storage and aging, which is opposite to the formation of free 5′-ribonucleotides, which initially peak and then decrease in a relatively short period of time—exposing a conflict of obtaining high amounts of both free amino acids and free 5′-ribonucleotides at the same time.

## 6. Food Safety

### 6.1. Biogenic Amines

An increase in ammonia and drip loss have been correlated with the decrease in overall quality rating and increase in the quality index score of squid and cuttlefish, with the TMA content markedly increased after 10 and 8 days of storage in squid and cuttlefish, respectively [39]. The food safety of squid can be managed by high-pressure processing (HPP). Specifically, HPP has been proven to lower the formation of autolytic activity and level of TMA [106]. HPP was also investigated for octopus, showing that TMA and DMA (dimethylamine) produced in chopped raw octopus treated at 600 MPa were significantly reduced by 42.5% and 62.2%, respectively, as compared with the levels in the control. The production of biogenic amines (BAs) increased by up to 1.82 mg/g in the control after 12 days of refrigerated storage, while the BA levels in the 450 and 600 MPa treated octopus were 1.40 and 1.35 mg/g, respectively [44].

### 6.2. Trace Elements

The concentrations of essential and non-essential elements in cephalopods have been determined to be within the prescribed limits set by various authorities, except for Cu and As. Regarding Hg and Pb intake, the consumption of cephalopods is not a cause of concern, and for the essential elements, cephalopod mollusc consumption makes an important contribution to the daily dietary intake of Cu, Zn, and Se. However, considerably higher contents are found in the hepatopancreas (i.e., the liver) than in the muscle of cephalopods, except for Hg and As, which are equally distributed in the two types of tissue. In particular, Cd present in the hepatopancreas is of dietary concern [35], which also seems to be the case, comparing tissue with digestive glands from other marine species, e.g., in the case of crustaceans [107]. It is therefore strongly recommended that the liver of cephalopod be not regularly consumed although it is showing a great umami taste potential in other investigations [5].

## 7. Cephalopod Gastronomy and Gastrophysics

### 7.1. Cephalopod Cooking and Cookbooks

Although different types of cephalopods have been used in food cultures around the world, there is no particular, well-defined universal cephalopod cuisine or cephalopod gastronomy [10]. Moreover, there are surprisingly few modern cookbooks specifically devoted to cephalopods [11,108,109], and they are mostly dealing with squid. In older accounts, octopus seems to be a favourite dish. The Roman gourmet Marcus Gavius Apicius (25 BCE−37 CE) is credited for the oldest known cookbook *De re coquinaria* from the Antique, which contains a recipe for octopus with pepper, lovage, ginger, and the Roman fish sauce garum [110]. From the Middle Ages, there are some recipes in handwritten manuscripts from the fourteenth century [111], e.g., a Catalan recipe for octopus filled with its own arms together with spices, parsley, garlic, raisins, and onions, and prepared over an open fire or in an oven. From the sixteenth century, a Catalan recipe describes the preparation of baked octopus and an Italian one using boiled, roasted, and marinated octopus [111].

While the preferred taste and texture of prepared cephalopods are very dependent on the food culture, there are some general trends. For example, Japanese eaters prefer a mild flavour as close to the natural flavour as possible, and they will therefore only add subtle flavours, e.g., from marinating liquids. Moreover, Japanese cuisine often uses squid that is either raw or extremely lightly cooked. In other places in Southeast Asia such as Vietnam, Thailand, and China, eaters prefer cephalopods with more spicy, powerful, and fishy flavours. In Southern Europe, cephalopods are almost always prepared either by grilling, steaming, or frying. In South America, marinating squid in salt and acid juices a la ceviche strikes a middle balance between raw and prepared/cooked.

Generally, the taste and flavour of fresh cephalopods are reasonably mild and quite easily blend in with other flavours. Therefore, in gastronomic uses of cephalopods, one may risk suppressing the subtle flavours of the cephalopods with stronger tasting ingredients, such that one is left with the texture as the only surviving characteristic of the cephalopod used.

Building on traditional, in particular Mediterranean cuisines, there appears in Europe to be an increasing interest among chefs to explore cephalopod cooking and take it into a more modernist setting [112]. This trend may be fuelled by a search for using marine food sources in a more sustainable fashion [11]. A deeper understanding of the gastrophysical properties of cephalopod muscular structure as reviewed in the present paper may aid in this search.

### 7.2. Culinary Gastrophysical Investigations of Cephalopods

A majority of scientific investigations into cephalopods include studies of raw tissue rather than cooked. Investigated cooking techniques of cephalopods have most frequently included boiling [59,70,81,82] and drying [45,67]. Other types of culinary preparations have involved tenderisation techniques of cephalopods using natural autolysis [74]; ultrasonic treatment [65]; and marination with NaCl and acetic acid [81], alkaline soaking [51], liver extract [73], collagenase [63], and spleen and bromelain extract [113]. However, it is often a predefined cooking treatment that is applied to study the physical, chemical, or microbiological changes rather than applying a gastrophysical approach to optimise the culinary treatment in question. Gastrophysical exploration of cephalopods is therefore limited, but a few examples include the investigation of *L. forbesii* [32,54,114]. Other studies involving cephalopods in a culinary context investigated (commercial) food preparations, ranging from frankfurters prepared from the mantle of *Dosidicus gigas* [66], ready-to-eat Indian squid (*L. dauvacelli* Orbigny) masala [76], and battered and fried squid rings [47].

The present authors have, in recent years together with several collaborators, including scientists, chefs, and innovators, developed a multipronged gastrophysical research programme with a focus on a single species, *L. forbesii,* that is common in Nordic waters but not part of the traditional Nordic food culture. Part of the motivation for undertaking such a programme can be seen as a contribution to the so-called New Nordic Cuisine. Another part pertains to the current search for sustainable food to promote a transition towards a more green and sustainable eating behaviour [115]. In our work, some of which is reported in the present review in a more general context, we tried to cover all aspects of squid food science and gastronomy, basically from the state of molecular structure, materials properties and chemical characteristics, preparation procedures, taste, flavour, and texture, to human food preferences and culinary innovation. Currently, this broad programme is being supplemented by an ethical dimension via participation in an international research programme dealing with humane slaughtering methods of cephalopods.

## 8. Outlook

In the future, there will be an increasing demand to replace the proteins from land-based animal food production that need to be scaled down to establish more sustainable global food production systems [1]. This has put focus on plant-based protein systems as well as little or poorly used marine sources [115]. Cephalopod populations have been on the rise over the last 60 years [8], and it is worthwhile to explore if certain cephalopods can be a new source of animal protein for human consumption. Among other studies, this will require research into the culinary science and gastrophysics of cephalopods as reviewed in the present paper.

Introducing new sources of marine protein may be a challenge for consumers who are unfamiliar with the eating textures that these types of little-known marine proteins of squid, cuttlefish, and octopus provide. Insight into how the eating texture of cephalopods can be managed, being structurally very different from what we know from protein structures in meat from terrestrial animals, is necessary to develop successful culinary preparation techniques, meal servings, and food products from cephalopods. Moreover, the special flavours from cephalopods must be studied and quantitatively accessed to optimise the eating quality. A key flavour attribute is the umami taste, which we have shown to be rich in squid and possibly also in other cephalopods. It has been proposed that the umami taste will become a driver in the green transition making more plant-based foods acceptable to a larger part of the population [11,115].

## Figures and Tables

**Figure 1 foods-11-02559-f001:**
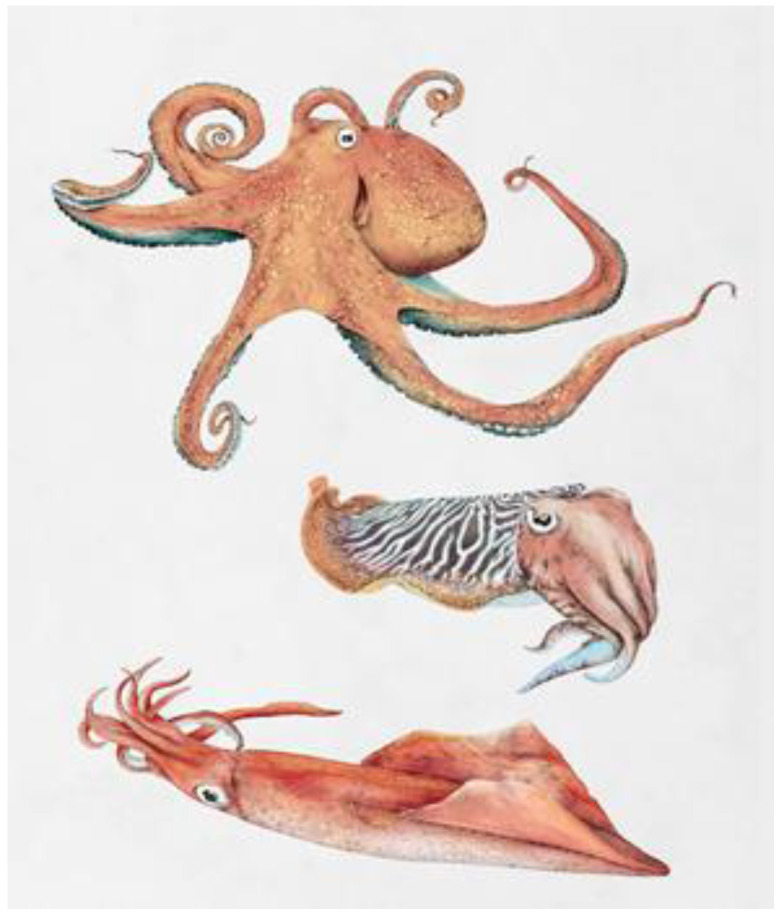
Cephalopods: (from the top) octopus, cuttlefish, and squid. Illustration by Ene Es.

**Table 1 foods-11-02559-t001:** Overview table of Latin and common names of some common species of cephalopods.

Cephalopod	Latin Name		Common Name	Origin
Family	Genus	Species		
Squid	*Doryteuthis*	*bleekeri*	Spear squid	Western Pacific Ocean
	*Dosidicus*	*gigas*	Jumbo squid	Eastern Pacific Ocean
	*Gonatopsis*	*boralis*	Boreopacific arm hook squid	North Pacific Ocean
	*Illex*	*argentinus*	Argentine squid	Southwestern Atlantic Ocean
		*coindetii*	Southern shortfin squid or broadtail shortfin squid	Mediterranean Sea
		*illecebrosus*	Northern shortfin squid	Northwest Atlantic Ocean
	*Loligo*	*bleekeri*	Spear squid	Western Pacific Ocean
		*chinensis*	Mitre squid	China Sea
		*duvauceli*	Indian squid	Mediterranean Sea
		*edulis*	Sword tip squid	Indian Ocean, The Arabian Sea
		*forbesii*	Veined squid and long-finned squid	North-east Atlantic Ocean
		*guncula brevis*	Atlantic brief squid	Atlantic Ocean
		*opalescens*	California market squid	Pacific Ocean
		*pealei*	Longfin inshore squid	Atlantic Ocean
		*vulgaris*	European squid and common squid	North Sea and South
	*Heterololigo*	*bleekeri*	Arrow squid, spear squid	Western Pacific Ocean
	*Ommastrephes*	*bartramii*	Neon flying squid	Northwest Pacific Ocean
	*Sepiteuthis*	*lesoniana*	Oval squid, bigfin reef squid	Indo-Pacific Ocean
	*Todarodes*	*pacificus*	Japanese flying squid, Japanese common squid, or Pacific flying squid	Northwest Pacific Ocean, Sea of Japan, Yellow Sea, and East China Sea
		*pacificus steenstrup*	Japanese flying squid	Western Pacific, excluding the Bering Sea; northern and eastern Pacific
		*sagittatus*	Sea arrow sagittate squid, red squid, European flying squid	Eastern Atlantic Ocean and the Mediterranean Sea
Octopus	*Eledone*	*moschata*	White octopus and musky octopus	Mediterranean Sea
		*cirrhosa*	Horned octopus, lesser octopus, and northern octopus	Northeast Atlantic
	*Octopus*	*vulgaris*	Pulpo, common octopus	Atlantic Ocean (Andalusia)
		*salutii*	Spider octopus	Mediterranean Sea and northeast Atlantic
Cuttlefish	*Sepia*	*aculeata*	Needle cuttlefish	Indian and Pacific Ocean
		*elegans*	Elegant cuttlefish	Eastern Atlantic, Mediterranean Sea including the Adriatic Sea, West Africa, and Agulhas Bank
		*officinalis*	Common cuttlefish and European common cuttlefish	Eastern Atlantic, Mediterranean Sea
		*orbignyana*	Pink cuttlefish	Atlantic Sea, Mediterranean Sea
		*pharaonis*	Pharaoh cuttlefish	Gulf and the Andaman Sea

Sources: [17,18,19].

**Table 2 foods-11-02559-t002:** Overview table of applied TA test types applied to analysing the mechanical texture of different species of cephalopods.

Family	Species	Cut	Test Type	Parameters	Fixture	Reference
Squid	*D. gigas*	Fins (gel)	Compression	Stress and strain	Cylindrical probe (dia. 38 mm), deformation 75%, load cell 100 N	[64]
	*D. gigas*	Mantle	Double compression	Flexibility, firmness	Cylindrical probe (dia. 5 mm)	[65]
	*D. gigas*	Mantle (cooked)	Double compression	Hardness, springiness, cohesiveness; shear force	Cylindrical probe (dia. 50 mm), deformation 75%; Warner–Bratzler shear blade	[66]
	*D. gigas*	Mantle (dried)		Firmness	Cylindrical probe (dia. 3 mm)	[67]
	*D. gigas*	Mantle (gel)	Double bite	Hardness, cohesiveness, elasticity	Knife blade, load cell 100 N	[68]
	*D. gigas*	Mantle (gel)	Double compression	Gel strength, elasticity, cohesiveness	Cylindrical probe, deformation 75%	[69]
	*D. gigas*	Mantle, fins, arms (cooked)	Shear force	Shear force		[70]
	*I. argentinus*	Mantle	Shear force	Shear force		[71]
	*I. argentinus*	Mantle (cooked)		Hardness, elasticity, chewiness		[72]
	*I. argentinus*	Mantle (raw)	Shear force		Wedge plunger	[33]
	*I. argentinus*	Mantle (raw, enzymes)		Toughness		[73]
	*I. coindetii*	Mantle	Tensile	Stress and strain	Two hooks	[74]
	*I. illecebrosus, L. pealei*	Mantle (raw, cooked)	Shear force		Modified Kramer shear cell	[75]
	*L- vulgaris*	Mantle (raw)	Compression test	Toughness	Cylindrical probe (5 kg load cell)	[36]
	*L. duvauceli*	Mantle (cooked)	Double compression	Hardness 1, hardness 2, cohesiveness, gumminess, springiness, chewiness; shear force	Cylindrical probe (dia. 50 mm), 50 N load cell, deformation 40%; Warner–Bratzler shear blade, 50 N load cell	[76]
	*L. duvauceli*	Mantle (raw, cooked)	Double compression	Cohesiveness, springiness, stiffness	Cylindrical probe (dia. 50 mm), deformation 40%	[59]
	*L. duvauceli*		Shear force	Shear force	Warner–Bratzler shear blade	[39]
	*L. edulis, I. argentinus*	Mantle (raw)	Shear force	Toughness	Plunger knife blade	[77]
	*L. forbesii, L. vulgaris*	Mantle (raw, cooked)	Shear force	Hardness, work	Warner–Bratzler shear blade	[32,54]
	*L. pealei*	Mantle (cooked)	Shear force	Force, energy	Single blade; punch and die	[28]
	*L. pealei, I. illecebrosus*	Mantle (raw, cooked)	Tension	Stress and strain	Tensile grips	[31,63]
	*L. vulgaris*	Mantle (raw)	Penetration	Firmness, elasticity, work	Flat bottom stainless-steel cylinder (dia. 6 mm) 100 kg load cell.	[78]
	*L. vulgaris*	Mantle (raw)	Texture profile analysis	Hardness, cohesiveness, springiness, gumminess, chewiness	Cylindrical compression probe	[36]
	*L. vulgaris*	Mantle (raw, enzymes)	Compression test	Toughness	Cylindrical probe using 40% compression (5 kg load cell)	[79]
	*O. sloanipacifcus*	Mantle (raw, dried)	Rupture	Shear force	Razor blade	[51]
	*S. lessoniana*	Mantle (cooked)	Tension	Stress and strain		[49]
	*T. pacificus*	Mantle (raw)	Compression	Hardness	Cylindrical probe (dia. 2 mm)	[42]
	*T. sagittatus*	Mantle (cooked)	Shear force	Toughness	Warner–Bratzler shear blade	[80]
Octopus	*E. moschata*	Mantle	Penetration	Toughness	Cylindrical probe (dia. 2 mm)	[81]
	*O. vulgaris*	Mantle	Penetration	Toughness	Cylindrical probe (dia. 2 mm)	[82]
	*O. vulgaris*	Arm (raw)	Texture profile analysis	Hardness, cohesiveness, springiness, gumminess, chewiness	Cylindrical compression probe	[36]
	*O. vulgaris*	Arm (raw)	Compression test	Toughness	Cylindrical probe (5 kg load cell)	[36]
Cuttlefish	*S. aculeata*	Mantle (raw)	Shear force	Shear force	Warner–Bratzler shear blade	[39]
	*S. officinalis*	Mantle (raw)	Texture profile analysis	Hardness, cohesiveness, springiness, gumminess, chewiness	Cylindrical compression probe	[36]
	*S. officinalis*	Mantle (raw)	Compression test	Toughness	Cylindrical probe (5 kg load cell)	[36]
	*S. pharaonis*	Mantle (raw)	Shear force	Toughness	Plunger knife blade	[77]

**Table 3 foods-11-02559-t003:** Overview table of umami taste compounds in different species of cephalopods and other marine sources of protein. EUC denotes effective umami concentration [87] that describes the synergy of simultaneous presence of free glutamate and free nucleotides in the sample.

Family		Species	Cut	Glutamate	Nucleotides	Nucleotides Identified	EUC	Reference
				mg/100 g	mg/100 g			
Cephalopod	Squid	*L. forbesii*	Mantle	110	24	AMP, IMP, XMP, GMP	3300	[5]
			Fins	72	26		2350	
			Arms	101	18		2300	
			Liver	462	39		22,400	
		*L. bleekeri*	Mantle	8	52	AMP	500	[88]
			Liver	90	12	AMP, IMP	1400	
		*L. edulis*	Mantle	16	45	AMP, IMP	900	[88]
			Liver	91	8	AMP, IMP	980	
		*S. lessoniana*	Mantle	4	44	AMP	220	[88]
			Liver	25	32	AMP, IMP	1000	
		*T. pacificus*	Mantle	7	40	AMP, IMP	350	[88]
			Liver	80	23	AMP, IMP	2320	
		*T. pacificus*		40	19	AMP, GMP, IMP	970	[89]
		*T. pacificus*		17	-	AMP, IMP	-	[90]
		*G. boralis*	Mantle	11	5	AMP	80	[91]
			Arms	28	7	AMP	270	
		*I. argentinus*	Mantle	170	3	AMP, IMP	790	[92]
		*O. bartrami*	Mantle	62	2	AMP, IMP	210	[92]
		*S. lessoniana*		4			-	[86]
		*S. lessoniana*		21	-	AMP, IMP	-	[90]
		*Heterololigo bleekeri*		28	-	AMP, IMP	-	[90]
	Octopus	*-*		25 *	90 *	AMP		[85]
	Cuttlefish	*-*		-	-			
Bivalve	Oysters	*C. gigas*	Whole body	160	44	AMP, GMP, IMP	8740	[6]
		*O. edulis*	Whole body	257	88	AMP, GMP, IMP	27,800	[6]
		*C. gigas*		145	26	AMP, IMP, GMP	4740	[93]
	Clam	*Paratapes undulatus*		90	5	AMP	640	[94]
	Mussel	*Mytilis edulis* L.	Female specimen, mantle	108	18	AMP, GMP, IMP, XMP	2480	[95]
			Female specimen, adductor muscle	233	31		9030	
			Male specimen, mantle	335	23		9720	
			Male specimen, adductor muscle	94	22		2560	
	Scallop	*-*		140	31	AMP, GMP	5430	[96]
Crustaceans	Shrimp	*Litopenaeus vannamei*		20	16	AMP	410	[97]
	Lobster	*Nephrops novergicus* L.		31	-		-	[98]
	Crab	*Chionoecetes opilio*		19	20	IMP, GMP, AMP	480	[96]
Teleost	Tuna	*-*			287	IMP, AMP	-	[96]
Anguillidae	Eel	*Anguilla anguilla*		22	-		-	[99]
>Algae	>Kelp	> *Saccharina japonica*	>	>1608	>-	>	>-	>[96]

Nucleotides are presented in IMP equivalence for ease of comparison, as calculated based on relative taste intensity values from [87]. * Values are the average of a presented interval. “-“ denotes that no data were available.

## Data Availability

Data is contained within the article.

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
