# Peer review of "Cephalopods as Challenging and Promising Blue Foods: Structure, Taste, and Culinary Highlights and Applications"

_foods, 2022, doi:10.3390/foods11172559_

Round 1

Reviewer 1 Report

the article indicates the links between the physicochemical and biological characteristics of a species with its characteristics and its gastronomic uses. Without revealing particularly innovative information, the publication draws up an exhaustive list of the latest data which underlines the interest of the consumption of cephalopods. The compilation of different sources makes it possible to classify the taxa according to their qualitative value as food. The publication also brings together data on the impact of different culinary uses on the gastronomic characteristics of products.

Nevertheless, it is difficult to find the bibliographical sources since they are presented by name of 1st author in the text and numbered in order of appearance in the bibliography. Homogenizing the presentation of bibliographical references would facilitate reading

Author Response

Reviewer 1

the article indicates the links between the physicochemical and biological characteristics of a species with its characteristics and its gastronomic uses. Without revealing particularly innovative information, the publication draws up an exhaustive list of the latest data which underlines the interest of the consumption of cephalopods. The compilation of different sources makes it possible to classify the taxa according to their qualitative value as food. The publication also brings together data on the impact of different culinary uses on the gastronomic characteristics of products.

Nevertheless, it is difficult to find the bibliographical sources since they are presented by name of 1st author in the text and numbered in order of appearance in the bibliography. Homogenizing the presentation of bibliographical references would facilitate reading.

Authors: We have fixed the reference list in accordance with Foods’ syntax.

Reviewer 2 Report

1.      Page 3, Actual pictures using a camera is better for the Figure 1.

2.      Page 4, Authors should review the relationship between “Cephalopod structure by biological constraints” and “food processing”.

3.      Page 10, Amino acids composition analysis is very important for food materials, authors should review more references in the part “3.3.2 Amino acids”.

4.      Line 494, Authors have reviewed amino acids and 5’-ribonucleotides in “5.1.1 Umami taste compounds”. Why are repetitive here? Soluble sugars are also important taste compounds.

5.      Authors should added more tables and figures.

6.      This manuscript needs English editing.

Author Response

Reviewer 2

  1. Page 3, Actual pictures using a camera is better for the Figure 1.

Authors: We would like to disagree. The hand-drawn illustrations were chosen to me more generic for the three groups of cephalopods described. Photographs would be species specific.

  1. Page 4, Authors should review the relationship between “Cephalopod structure by biological constraints” and “food processing”. 

Authors: We believe that this is what we have actually done in the paper seen as a whole. We have in section 2 displayed biological constraints on pre-processing (freeze storage effects), please see section 2.2.3. Effects on food processing (i.e., all processing after handling and storage) will be much dependent on the actual food preparation in question, and the reader is advised to the appropriate information in the specific literature, of which we have presented a selection in section 7.2. We have in included a summarizing statement at the beginning of section 2. to make this clearer.

  1. Page 10, Amino acids composition analysis is very important for food materials, authors should review more references in the part “3.3.2 Amino acids”.

Authors: This is certainly true and that is why we have devoted a separate section 5.1.1 to analysis of amino acids in the context of taste and a section of their degradation during food processing in section 5.2. We have modified the text in the beginning of sections 3.3 and 3.3.2 to make this clear.

  1. Line 494, Authors have reviewed amino acids and 5’-ribonucleotides in “5.1.1 Umami taste compounds”. Why are repetitive here? Soluble sugars are also important taste compounds.

Authors: This layout of the paper was chosen in a way to be as logic as possible. In fact, to minimize overlap, description of free amino acid analysis naturally belonging in section 3.3 was placed together with nucleotides in 5.1.1 since the focus of the paper is on taste. See also our response to point 3. above.

  1. Authors should add more tables and figures.

Authors: We have in the review included and reviewed all the material we found relevant and available in the literature. The reviewer seems no to have noted any shortcomings in the material presented in the review. Hence, we see no reasons to include additional figures and tables for the mere reason just to make the paper longer.

  1. This manuscript needs English editing.

       Authors: We have performed a thorough proofreading and editing of the English language.

Round 2

Reviewer 2 Report

1. Page 10, 3.3.2 Amino acids. Authors should focus on the “Nutritional value of protein” by amino acid composition analysis, not only free amino acids (taste).

2. Authors have reviewed amino acids and 5’-ribonucleotides in 5.1.1, but they are repetitive in 5.1.2. Authors should focus on other taste compounds in 5.1.2.

Author Response

Authors response to Reviewer #2 revised version review comments

We thank the reviewer for the comments on the revised manuscript. Below in italics we address the comments.

Comments by Reviewer #2 for the revised version:

  1. Page 10, 3.3.2 Amino acids. Authors should focus on the “Nutritional value of protein” by amino acid composition analysis, not only free amino acids (taste).

We respectfully wish to disagree with the reviewer. This paper is not concerned with the nutritional value of cephalopods but their gastronomic potential. Hence we have focussed on the taste of free nucleotides and nucleotides.

  1. Authors have reviewed amino acids and 5’-ribonucleotides in 5.1.1, but they are repetitive in 5.1.2. Authors should focus on other taste compounds in 5.1.2.

We addressed this point in our previuous rebuttal letter. We here copy our response: “Authors: This layout of the paper was chosen in a way to be as logic as possible. In fact, to minimize overlap, description of free amino acid analysis naturally belonging in section 3.3 was placed together with nucleotides in 5.1.1 since the focus of the paper is on taste. See also our response to point 3. above.” There is no unique way of making the perfect layout of a complex reveiw paper of the present type but we believe that we have struck a proper balance between minimal overlap and maximal readability (and hence service to the readers). The reviewer asks us to also focus on ‘other taste compounds’. We have focussed on umami taste compounds for the reasons given in the introduction of the paper. Including other taste and flavour compounds would be a major undertaking and lead to quite a different (and much longer) paper.

Hence we have found no reasons to further revise the paper.
